Giants in the landscape: status, genetic diversity, habitat suitability and conservation implications for a fragmented Asian elephant (Elephas maximus) population in Cambodia

Sinovas Pablo 1 pablosinovas@hotmail.com
http://orcid.org/0000-0003-1178-5432 Smith Chelsea 2
Keath Sophorn 1 3
Chantha Nasak 1
Kaden Jennifer 4
Ith Saveng 1 3
http://orcid.org/0000-0003-1186-2717 Ball Alex 4
1 Fauna & Flora , Phnom Penh , Cambodia
2 Fauna & Flora , Cambridge , United Kingdom
3 Department of Biology, The Royal University of Phnom Penh , Phnom Penh , Cambodia
4 RZSS WildGenes, Royal Zoological Society of Scotland , Edinburgh , United Kingdom
McElligott Alan
Electronic publication date: 2025 Mar 13
Publication date: 2025
Volume: 13
Electronic Location ID: e18932
Received 2024 Feb 21; Accepted 2025 Jan 14
Copyright: © 2025 Sinovas et al.
Copyright year: 2025
Copyright holder: Sinovas et al.
License: This is an open access article distributed under the terms of the Creative Commons Attribution License, which permits unrestricted use, distribution, reproduction and adaptation in any medium and for any purpose provided that it is properly attributed. For attribution, the original author(s), title, publication source (PeerJ) and either DOI or URL of the article must be cited.
License URL: https://creativecommons.org/licenses/by/4.0/

Keywords: Genetic diversity, Asian elephant, Elephas maximus, Cambodia, Southeast Asia, Population assessment, Habitat suitability, Habitat modelling, Conservation, Fragmentation

Funding: USAID Greening Prey Lang Project People’s Postcode Lottery This work was supported by the USAID Greening Prey Lang project and the People’s Postcode Lottery. The funders had no role in study design, data collection and analysis, decision to publish, or preparation of the manuscript.

==============================
Asian elephant (Elephas maximus) populations are declining and increasingly fragmented across their range. In Cambodia, the Prey Lang Extended Landscape (PLEL) represents a vast expanse of lowland evergreen and semi-evergreen forest with potential to support Asian elephant population recovery in the country. To inform effective landscape-level conservation planning, this study provides the first robust population size estimate for Asian elephants in PLEL, based on non-invasive genetic sampling during the 2020–2021 dry season in three protected areas: Prey Lang, Preah Roka and Chhaeb Wildlife Sanctuaries. Further, it provides an assessment of the species’ range, habitat suitability and connectivity within the landscape using Maxent and Fuzzy suitability models. Thirty-five unique genotypes (individual elephants) were identified, of which six were detected in both Preah Roka and Chhaeb Wildlife Sanctuaries, providing evidence that elephants move readily between these neighbouring protected areas. However, no unique genotypes were shared between Preah Roka/Chhaeb and the less functionally connected southerly Prey Lang Wildlife Sanctuary. The estimated population size in the southern population was 31 (95% CI [24–41]) individuals. The northern population of Preah Roka/Chhaeb Wildlife Sanctuaries is estimated to number 20 (95% CI [13–22]) individuals. Habitat loss is prevalent across the landscape and connectivity outside of the protected areas is very limited; however, large swathes of suitable elephant habitat remain. As the landscape holds the potential to be restored to a national stronghold for this flagship species, in turn resulting in the protection of a vast array of biodiversity, we recommend protection of remaining suitable habitat and reduction of threats and disturbance to elephants within these areas as top priorities. Our study offers a model for integrated elephant population and landscape-level habitat modelling that can serve to guide similar research and management efforts in other landscapes.

Introduction

Global biodiversity is decreasing at alarming rates, with the most rapid declines occurring in the tropics (Bradshaw, Sodhi & Brook, 2009). This biodiversity crisis is particularly acute in Southeast Asia (Sodhi et al., 2009; Estoque et al., 2019), where large swathes of land have been subject to defaunation, largely driven by habitat alteration and hunting (Tilker et al., 2019). Defaunation, the decline or extinction of animal populations or species, is a major driver of global ecological change (Dirzo et al., 2014). It can have profound cascading ecological effects, ranging from co-extinctions of interacting species to declines in genetic and functional diversity and to the loss of ecosystem services (Young et al., 2016). A robust scientific understanding of the ecology and conservation status of animal species is required to slow defaunation, yet ecological knowledge of fauna is particularly lacking in the tropics (Young et al., 2016). For example, the most understudied large herbivores, such as Asian elephants (Elephas maximus) and gaur (Bos gaurus), occur in developing nations; considering the crucial role that this guild plays in the ecosystem, from acting as ecosystem engineers to promoting seed dispersal and nutrient cycling, urgent action is needed to address research gaps and management needs for large herbivores (Ripple et al., 2015). More broadly, an integrated strategy, underpinned by landscape-level conservation planning is required to reverse the trend of terrestrial biodiversity loss (Leclère et al., 2020).

The Asian elephant is categorised as Endangered with a decreasing population trend on the IUCN Red List (Williams et al., 2020), illustrating the trend and threats faced by much of the world’s biodiversity, including large herbivores (Ripple et al., 2015). The species is threatened primarily by widespread habitat loss and fragmentation (Williams et al., 2020), particularly in Southeast Asia (Leimgruber et al., 2003). As the largest land mammal on the Asian continent, it not only requires species-level monitoring and conservation action but also landscape-level management, thus necessitating an understanding of both population parameters and the suitability, use and fragmentation of its habitat within the landscape (Rood, Ganie & Nijman, 2010). Research on the genetic diversity and population structure of Asian elephants across their range has been highlighted as a particular need to inform conservation strategies and to help identify populations that require targeted conservation efforts (Zakaria et al., 2024).

Calls for prioritisation of conservation efforts in Southeast Asian populations have been getting louder (Budd et al., 2023), and Cambodian populations remain particularly understudied. The Cambodian lowlands encapsulate the pressures faced by landscapes across Southeast Asia and the tropics more widely, including deforestation and illegal hunting (Lohani et al., 2020; Gray & Gauntlett, 2019). Fragmentation of the elephant populations in Cambodia is particularly acute, where fewer than 600 Asian elephant individuals are thought to remain. The core sub-populations are found in the Cardamom Mountains in the southwest and Eastern Plains to the east (Maltby & Bourchier, 2011). Smaller sub-populations persist in other areas of the country, including within the Prey Lang Extended Landscape (PLEL) (Maltby & Bourchier, 2011), a large expanse of lowlands in central and northern Cambodia covering approximately a sixth of the country’s surface, between the Mekong river to the east and Tonle Sap lake to the west.

The PLEL is extremely understudied with very little known about its remaining elephants. A robust understanding of the remaining population in the PLEL is necessary to design effective management interventions to support and monitor population recovery. Increased knowledge could also leverage elephants as a flagship species to the benefit of overall landscape-level biodiversity conservation in the region. Asian elephants are generally recognized as flagship species (e.g., Blake & Hedges, 2004; Barua, Tamuly & Ahmed, 2010). Flagship species can capture the attention of stakeholders and the public to generate support for positive conservation outcomes, for example by creating a moral or cultural imperative for policy, promoting inter-institutional planning, or justifying site protection (Jepson & Barua, 2015). However, there are currently no reliable estimates of elephant population size in the PLEL, nor studies of habitat suitability and fragmentation in the landscape.

Population size is a key component required for the assessment of conservation actions (Callaghan et al., 2024). Being able to effectively determine whether conservation interventions are working often requires robust estimates of population sizes before and after the intervention (Nuttall et al., 2021). Population size is also vital in predicting the viability of the population and deciding where limited conservation resources are targeted (Saunders, Cuthbert & Zipkin, 2018). For the elusive Asian elephant, it has long been acknowledged that more accurate population size estimates are needed for informed decision making (Blake & Hedges, 2004). However, we acknowledge that other demographic components are also required for robust assessments of population viability, especially for long-lived species, as elephants can persist for many years in sub-optimal habitat without recruitment of the next generation (Armbruster, Fernando & Lande, 1999; De Silva & Leimgruber, 2019). It is therefore also important to understand the demography of the population, including the age and sex of individuals. The genetic diversity of the population is also paramount for long-term viability, as fragmented populations are at greater risk of inbreeding; the negative consequences of which can be masked in long-lived species (Taylor et al., 2017). We therefore aim to use a multifaceted approach to understanding the remaining PLEL population.

The main aim of this study is to combine field surveys, genetic techniques and habitat modelling to understand the Asian elephant population demography and habitat use in PLEL for the first time. In this study, we used a non-invasive genetic sampling survey to assess population size, the sex ratio and genetic diversity. We also used modelling approaches to identify areas of suitable habitat and key connectivity corridors for the species. Together, these methods aim to inform evidence-based decision making on the management of one of the largest remaining lowland forests in Cambodia. Additionally, this methodology may provide a blueprint for future integrated assessments of Asian elephants and other umbrella species to inform wildlife and protected area management policy and practice in Cambodia and beyond.

Materials and Methods

Study area

We focused our study on the Asian elephant sub-populations within the Prey Lang Extended Landscape (PLEL), a large expanse of mostly flat land in central-northern Cambodia west of the Mekong river. The PLEL is characterised by a mosaic of lowland evergreen forests and agricultural land, with elevations below 200 m above sea level. Its climate is tropical, characterised by a wet season from May to November and a dry season from December to April. Within PLEL, the study area (103°57′–105°58′ E and 14°26′–12°34′ N) comprises three protected areas where elephant presence has been recorded over the past decade (Prey Lang Wildlife Sanctuary, Preah Roka Wildlife Sanctuary and Chhaeb Wildlife Sanctuary), as well as one protected area (Kulen Promtep Wildlife Sanctuary) where elephant presence has not been recorded over the past decade and was therefore not sampled (Fig. 1).

Figure 1 Area of interest for study, demonstrating the dung sample sites within the PLEL protected areas and additional protected areas in the region.

The area of interest was created from a minimum bounding polygon of the PLEL protected areas buffered by 10 km and clipped to the Cambodian border. Image source credits: Esri (2022), IUCN (2021) and UNEP-WCMC (2022).

Elephant presence

Available elephant occurrence records were collated from relevant stakeholders: Cambodia’s Ministry of Environment and conservation NGOs (Fauna & Flora International, Conservation International, Wildlife Conservation Society and Wild Earth Allies). We conducted non-invasive genetic sampling, involving the collection of dung samples, during the 2020–2021 dry season in Prey Lang, Preah Roka and Chhaeb Wildlife Sanctuaries (see below), in areas where elephant presence had been noted by local rangers. These surveys in turn generated additional elephant presence records for the target protected areas. All occurrence records were then used to inform the habitat suitability modelling.

Population survey

Our design followed survey standards for elephant monitoring through capture-recapture sampling (Hedges & Lawson, 2006; Karanth, Nichols & Hedges, 2012). Survey methods were non-invasive, relying on sample collection from elephant dung with the relevant permit from Cambodia’s Ministry of Environment (permit 248-MoE), with no contact made with elephants. Using faeces for DNA collection is in accordance with the 3Rs principle for animal welfare as it is non-invasive (Zemanova, 2020).

We identified focal areas for faecal collection based on data gathered from previous studies in the landscape (personal communication with locally-active conservation NGOs) and on key informant interviews with one staff from each of three conservation NGOs working in the area and with the head Ministry of Environment rangers in charge of patrolling each of the protected areas within the landscape. The study was government-sanctioned, including the interview phase (permit 248-MoE), and the purpose of the interviews was explained in advance to all interviewees, who participated voluntarily. Survey areas were 625 km2 in Preah Rokar/Chhaeb Wildlife Sanctuaries and 1,000 km2 in Prey Lang Wildlife Sanctuary.

Samples from elephant dung were collected during the 2020–2021 dry season between 29th November 2020 and 14th March 2021 by teams of trained field researchers. Five collection visits were conducted in each wildlife sanctuary and faecal samples were collected via two methods. The first method involved using a cotton swab to collect genetic material from the outside of a dung bolus; this has proved successful at preserving DNA collected from African forest elephant (Loxodonta cyclotis) dung samples (Bourgeois et al., 2019). The second method involved collecting exterior parts of a bolus and preserving in absolute ethanol; this was used as a backup, as the technique provides material for repeat extractions. The second method has previously proved successful for DNA collection from Asian elephant dung samples (Fernando et al., 2000). We followed the methods outlined in these aforementioned studies with minor alterations that can be found in Article S1.

DNA extraction and quality check

Samples were transported immediately after each field trip to the Royal University of Phnom Penh (RUPP) Conservation Genetics Laboratory and stored in a −20 °C freezer until DNA extraction. QIAamp® Fast DNA Stool Mini kits were used for DNA extraction with some initial preparation differences depending on whether the sample was a swab or dung preserved in ethanol (see Article S1 for details). A total of 199 dung piles were sampled, with 155 (78%) being sampled via both collection methods. Therefore, a total of 354 sample tubes were provided to the RUPP Conservation Genetics laboratory between Dec 2020–March 2021. DNA preservation success rates for each collection method were determined using 20 dung piles from which both a swab sample and an ethanol sample were obtained. Comparisons were made between the DNA extractions based on performance in one of the methods required for individual identification (Single Nucleotide Polymorphism-SNP-genotyping).

An attempt to genotype the elephant producing each sampled dung pile was made to individually identify the elephants within the PLEL and therefore provide recapture rates. Using faecal samples to estimate population sizes via a capture-recapture approach using genetic material was first performed in 1999 (Kohn et al., 1999) and has become a widely used tool within conservation, including for estimating elephant population size (Eggert, Eggert & Woodruff, 2003; Gray et al., 2014; Bourgeois et al., 2019). Here we use two genetic marker sets for genotyping, one based on microsatellites and the other on SNPs. These are both presumed neutral marker-sets distributed across the nuclear genome.

SNP genotyping

Twenty SNPs have previously been screened for use in Cambodian elephant populations. All SNPs were biallelic and identified based on high minor-allele frequencies. The genotyping method used KASP assays (LGC Genomics, Teddingto, Herts, UK), with fluorescence after PCR amplification measured on a StepOne™ Real-Time PCR System (Thermo Fisher Scientific, Waltham, MA, USA). DNA extracts were diluted 1:5 to decrease inhibition prior to amplification. Included on every 48-well plate were two negative controls and 2–3 positive controls (blood samples) with known variability in genotypes. Genotyping was only conducted after checking that the results of the controls were successful. The PCR protocol can be found in Article S1 and details of the KASP assays can be found in Table S1.

Microsatellite genotyping

DNA extracts were diluted 1:5 and genotyping was attempted at nine microsatellite markers in two panels. These microsatellites had been previously designed for use in elephants and have been used to study genetic diversity in Asian elephants (Kongrit et al., 2008; Comstock, Wasser & Ostrander, 2000; Nyakaana & Arctander, 1998). All microsatellite genotyping was carried out in triplicate and a consensus genotype was created for each sample if the allele was observed ≥2 times. A negative and positive control (blood sample) were included on each plate. See Article S1 and Table S2 for details of the microsatellites used in this study, including primer sequences, fluorescent tags and PCR protocols used for multiplexing.

Population structure

The nuclear markers (SNPs and microsatellites) were used to test for population structure in the Cambodian samples using three methods. Firstly, a Bayesian method using the program STRUCTURE (Pritchard, Stephens & Donnelly, 2000) was performed using an MCMC of 1,000,000, an initial 250,000 were run and discarded as burn-in. An admixture model (infer alpha) and correlated allele frequencies model (Lambda = 1) was used for a K of 1–5. All runs were performed in triplicate to verify the stability of the results. Secondly, a principle component analysis (PCA) was implemented using the dudi.pca command in the ‘adegenet’ package in R Studio (Jombart, 2008). Lastly, the fixation index (FST) was calculated using hierfstat (Goudet, 2005) using the Weir and Cockerham method (Weir & Cockerham, 1984), values were compared to the divergence levels described in the original method development (Wright, 1965; Hall, 2022). All three analyses were performed on the unique genotypes identified via the individual identification methods outlined below, however unique genotypes that were scored at <4 microsatellites were removed (n = 8), to reduce the amount of missingness. Any markers that showed evidence of null alleles, according to the program Cervus v3.0.7 (Kalinowski, Taper & Marshall, 2007), were also removed before analysis. We used three methods to reduce the risk of incorrect assessment, as multiple studies have highlighted assumptions and biases that can result from population genetic approaches (Puechmaille, 2016; Loog, 2021; Elhaik, 2022).

Mitochondrial sequencing

A 152bp mitochondrial sequence was produced using primers designed specifically for use in degraded Asian elephant faecal samples. Due to the degraded nature of DNA extracted from elephant faecal samples and the potential of amplifying non-target nuclear mitochondrial (NUMT) DNA, the mitochondrial sequence was created using two separate primer pairs that were each designed to amplify a small section of an overlapping region of the mitochondrial d-loop. The primers used in the PCR for each fragment were designed using Primer 3 (Untergasser et al., 2012) and a section of the Asian elephant mitochondrial genome sequence (Accession: DQ316068.1). The fragment 1 primers (AEL_dloop) were designed to amplify a 204bp sequence and the fragment two primers (CR_AEL_RZSS) to amplify a 193bp sequence, the overlapping region between them was 111bp. Stringent quality checks and end trimming was conducted in Geneious Prime v 2021.1.1, hence the final alignment being smaller than each target mitochondrial product. This target region was chosen as it is a highly variable section of the mitochondrial genome that exhibits variation between maternal lineages. This section of the control region also overlaps sequencing performed in Asian elephant samples from another eight countries (Vidya, Sukumar & Melnick, 2009), allowing comparisons to be made between the PLEL population and others. Details of the primers and PCR protocols are provided in Article S1 and Table S2. A haplotype network was created in the program PopArt v1.7 using the median joining network algorithm (Bandelt, Forster & Röhl, 1999).

Elephant sexing

The method developed by Ahlering et al. (2011) was used to sex each sample. This technique uses three primer sets to amplify regions of three genes on the sex chromosomes, 1 on the X chromosome (PLP) and 2 on the Y chromosome (AMELY and SRY). As female elephants are homogametic and male elephants are heterogametic, the three genes will only amplify in male elephants, with only one amplifying in females (PLP). The PCR amplification followed the protocol of Ahlering et al. (2011) and products were visualised on a 2% agarose gel stained with GelRed®. A negative and two positive controls (known male and female) were included on every PCR plate. Any samples that showed unclear band separation were tested a second time (n = 8).

Genetic diversity

Two approaches to measuring genetic diversity were used in this study, one using the mitochondrial data and the other using the nuclear microsatellite markers. Haplotype diversity (Hd) and nucleotide diversity of the mitochondrial sequences were calculated in DNASP v6.12.03 (Rozas et al., 2017). As this same mitochondrial region has been sequenced in elephants from other countries (Vidya, Sukumar & Melnick, 2009) a 152bp alignment was created using the geneious alignment algorithm in Geneious Prime v 2021.1.1 for diversity comparisons between 15 populations in nine countries. The nuclear diversity was calculated using only the individually identified elephants which were genotyped at ≥4 microsatellites. Only the microsatellites which did not show any evidence of null alleles were included. Observed heterozygosity (Ho), Expected heterozygosity (He), Rarified Allelic richness (Ar) and the Fixation index (FIS) were calculated in the R package hierfstat (Goudet, 2005). Given the notorious difficulty of standardising microsatellite genotyping between studies and laboratories (Hoffman & Amos, 2005; Moran et al., 2006; Ellis et al., 2011) comparisons of the results to other studies were not conducted.

Individual identification

The SNP and microsatellite genotyping data were combined to perform individual identification of each dung sample. Probability of identity scores (PID and PIDSIB) were calculated for the marker-sets in the program Cervus v 3.0.7 (Kalinowski, Taper & Marshall, 2007). Only a marker-set with a PIDSIB score of <0.01 was deemed acceptable for analysis. This led to the inclusion of dung samples that were either i) genotyped at ≥18 SNPs or ii) genotyped at ≥15 SNPs AND ≥2 microsatellites. Individual identification was then performed in the R package Allelematch (Galpern et al., 2012) using an allele mismatch score of 6. Any mismatches were manually checked and groups amalgamated into unique genotypes if they shared multiple matches where >50% of mismatches could be attributed to allelic dropout.

Population size estimates

Population size estimates were made from the unique genotypes generated by the genetic analysis results. The R package Capwire, based on the models presented by Miller, Joyce & Waits (2005), was utilised to estimate population size. Capwire has been used in a range of population estimation studies and is recognised within the field (Lonsinger et al., 2019; Davidson et al., 2014). The Two-Innate Rates Model (TIRM) model was run as it accounts for heterogeneity of capture probabilities for different individuals. The model assumes a closed population and the estimate is made from the capture class (the number of times individuals have been captured i times) and the frequency of each capture class.

Habitat suitability and connectivity modelling

A literature review of available studies identified forest cover (Lin et al., 2008; Gray et al., 2014), water accessibility (Sharma et al., 2020), and anthropogenic disturbance as key variables affecting habitat suitability (Gray & Phan, 2011; Huang et al., 2019; Sharma et al., 2020; Neupane et al., 2019). Data was sourced to represent these variables in suitability models: Global Forest Watch for forest cover and deforestation data (Hansen et al., 2013); Open Development Cambodia dataset for rivers (Open Development Cambodia, 2024); and Open Street Map data for roads and settlements (OpenStreetMap Contributors, 2024). Studies of Asian elephants in Cambodia are underrepresented in the academic literature, with a geographic bias towards China and India (Lin et al., 2008; Liu et al., 2016; Huang et al., 2019). In this study, we represent anthropogenic disturbance by road kernel density, Euclidean distance to villages, and recent and historic deforestation prepared in ArcGIS Pro v2.8 (Esri, 2024). Data sources and preparation can be found in the Supplemental Materials (Table S3).

MaxEnt

MaxEnt models were run to provide variable weighting importance for the weighted fuzzy habitat suitability models (Phillips, Anderson & Schapire, 2006). MaxEnt additionally provides insight into areas of suitable habitat by considering the relationships between variables and occurrence locations. Models were run using the default parameter settings on a 10-fold cross-validation model with a bias file included to minimise overfitting caused by sampling bias and a maximum of 5,000 iterations permitted (Kramer-Schadt et al., 2013; Syfert, Smith & Coomes, 2013). The bias file was created in SDMToolbox V2.4 (Brown, Bennett & French, 2017). Results were then evaluated using the Area Under Curve (AUC) to assess appropriateness for use in conservation management.

Fuzzy suitability model

A fuzzy habitat suitability model was used to incorporate local conservation expert opinions on the relationship, either positive or negative, between variables identified in the literature that affect habitat suitability including elephant relationships to forest cover, areas of deforestation, rivers, villages and road density. A fuzzy approach was taken to account for uncertainties in specific thresholds outlined by local experts for the variables in ranges in this understudied area (Ouellet et al., 2021). The model was run in ArcGIS Pro v2.8 using the Fuzzy Memberships tool (Esri, 2021) to apply linear memberships to all variables and the Weighted Overlay tool to combine layers to identify suitability. The application of linear memberships to variables enabled incorporation of uncertainty within expert opinions. MaxEnt results were then used to weight variables in accordance with their contribution to model performance as proxy for importance for elephant habitat suitability. Details on fuzzy memberships and weightings can be found in the Supplemental Materials (Table S3).

The total suitable habitat for Asian elephants was identified by aggregating the fuzzy and Maxent outputs. Fuzzy output areas of >5 were deemed suitable (Aini et al., 2017). Maxent outputs were reclassified using the maximum test specificity plus sensitivity threshold (Liu, Newell & White, 2016). The binary maps were then added to form a comprehensive map of habitat suitability.

Connectivity

Morphological Spatial Pattern Analysis (MSPA) was run using the prepared forest canopy layer to identify core areas of forest. A least-cost path model, adjusted to apply circuitscape theory to identify pinch points in connectivity, was run in Linkage mapper between core areas >25 km2 identified in MSPA. A resistance surface was created using Gnarly Landscape Utilities version 0.1.0 (Shirk & McRae, 2013), non-forest areas with <15% canopy cover, settlements and roads, were designated with the highest resistance scores (Table S4) (Suksavate, Duengkae & Chaiyes, 2019; de la Torre et al., 2019). As one of the first studies to analyse Asian elephant connectivity in PLEL the default parameter settings were used including a 200,000 cost weighted distance truncation factor.

Results

Population study

DNA extraction and quality check

Initially, DNA was extracted from 20 dung piles that had samples collected via both methods, dung in ethanol and a swab in Isohelix™ buccalfix. A significantly greater number of genotypes could be obtained for the DNA extracted from the swabs compared to the DNA extracted from the ethanol samples (Welch Two Sample t-test, t = −2.6071, df = 37.176, p-value = 0.01307). An average of 15/20 SNPs could be genotyped in swab samples compared to 9/20 in the ethanol samples (Fig. S1). All further DNA extraction, therefore, focused on the samples collected via the swabbing method; a total of 163/199 dung samples were sampled using a swab.

Nuclear marker genotyping (SNPs and microsatellites) and individual identification

A total of 106/163 (65%) of the samples were genotyped at ≥18 SNP markers. The microsatellite genotyping was less successful, a total of 60/163 (37%) of the samples were genotyped at ≥5 microsatellite markers. A combined dataset using both the SNP and microsatellite genotypes maximised the number of useable samples and provided the ability to identify unique genotypes. This produced a dataset of 112/163 (69%) samples with the markers having a combined PID exclusion value of 0.00000465 and a PIDSIB of 0.00211013. Therefore, each unique genotype is highly likely to represent a single individual elephant and this assumption was used for the Mark-recapture approach. A total of 35 unique genotypes (UG) were identified. The resampling rate varied widely between these individuals; one genotype was observed in 11 dung samples; however, 15 of the unique genotypes were only identified on a single occasion (see Table S5).

Six of the unique genotypes were isolated from dung samples in both Preah Roka Wildlife sanctuary and Chhaeb Wildlife sanctuary providing evidence that elephants are moving between these two neighbouring protected areas. All unique genotypes identified from dung samples in the Prey Lang Wildlife sanctuary were not observed in either of the other two areas. The raw microsatellite fragment analysis files and SNP fluorescent data have been uploaded to DRYAD here, DOI: 10.5061/dryad.m905qfv88, and here, DOI: 10.5061/dryad.ttdz08m5g, respectively.

Population structure

The Principal Component Analysis (PCA), the Bayesian STRUCTURE analysis and the fixation index (FST) suggest little to no genetic divergence between the elephants in Preah Roka/Chhaeb and the elephants in the Prey Lang Wildlife sanctuaries. Two markers (SNP023 & FH48) showed evidence of null alleles and FIS values >0.5, and were removed prior to analysis (Table 1). The structure results show no evidence of genetic differentiation between the two areas at K = 3 (Fig. 2 and Fig. S2). The PCA analysis shows a large overlap between the elephant genotypes in the north and south and the low eigen values only explain a small amount of variance, the eigen values produced for PC1 (0.896) and PC2 (0.796) explain 14.2% and 12.6% of the variation, respectively (Fig. 3). The FST value between the elephants found in Prey Lang and those in Preah Roka/Chhaeb was 0.066, indicating a moderate level of divergence. Although these results suggest no, to low divergence, the lack of evidence of any elephants moving between Prey Lang and the other two areas, and the long generation times of elephants, caused us to conduct the remaining analyses separately assuming two populations, Preah Roka/Chhaeb and Prey Lang.

Table 1 Summary statistics of the genetic markers (SNPs and microsatellites) genotyped in elephant samples collected in Preah Roka/Chhaeb and Prey Lang Wildlife sanctuaries.

Only unique genotypes as identified via Allele-match were included in the analysis and individuals genotyped at <4 microsatellites were excluded. All parameters calculated in Cervus v 3.0.7 except FIS which was calculated in Fstat v 2.9.3.

Locus ID	k	N	HObs	HExp	PIC	F(Null)	FIS	
SNP002	2	26	0.5	0.419	0.326	−0.0983	−0.194	
SNP004	2	25	0.24	0.216	0.189	−0.0588	−0.096	
SNP005	2	25	0.36	0.301	0.252	−0.0969	−0.292	
SNP010	2	26	0.615	0.462	0.35	−0.1524	−0.327	
SNP011	2	27	0.37	0.492	0.366	0.1318	0.242	
SNP013	2	27	0.37	0.307	0.256	−0.1002	−0.214	
SNP016	2	26	0.462	0.462	0.35	−0.0097	−0.003	
SNP021	2	27	0.111	0.107	0.099	−0.0194	−0.163	
SNP023	2	26	0.154	0.462	0.35	0.4927	0.67	
SNP025	2	27	0.222	0.307	0.256	0.1518	0.184	
SNP026	2	26	0.346	0.382	0.305	0.04	0.081	
SNP027	2	27	0.593	0.453	0.346	−0.1429	−0.299	
SNP029	2	27	0.111	0.107	0.099	−0.0194	−0.053	
SNP031	2	27	0.593	0.453	0.346	−0.1429	−0.308	
SNP033	2	26	0.577	0.491	0.366	−0.0901	−0.365	
SNP035	2	27	0.148	0.14	0.128	−0.0302	−0.087	
SNP036	2	27	0.37	0.352	0.286	−0.0345	−0.073	
SNP037	2	27	0.37	0.391	0.31	0.0182	0.03	
SNP039	2	27	0.296	0.307	0.256	0.0092	−0.02	
SNP040	2	27	0.296	0.307	0.256	0.0092	0.055	
Emu04	5	27	0.556	0.655	0.578	0.0797	0.035	
FH48	2	27	0.074	0.307	0.256	0.6033	0.657	
Emu07	6	26	0.615	0.821	0.777	0.1336	0.213	
Emu03	2	17	0.294	0.401	0.314	0.1392	0.279	
Emu12	3	27	0.481	0.398	0.352	−0.1271	−0.57	
Emu10	5	25	0.6	0.749	0.687	0.0981	0.17	
Emu17	5	14	1	0.786	0.719	−0.1481	−0.327	
Emu15	6	18	0.778	0.663	0.615	−0.114	−0.222	
Note:

K, number of alleles; N, individuals genotyped; HObs, Observed heterozygosity; HExp, Expected heterozygosity; PIC, Polymorphic information content; F(Null), Null allele frequency; FIS, fixation index as calculated in Weir & Cockerham (1984) for small f.

Figure 2 Population structure plot for the individual genotypes obtained from PLEL.

Results include a priori population sizes of k = 2 and k = 3 and individuals are ordered according to the protected area(s) they were found in, Prey Lang (PL) or Preah Roka/Chhaeb (PRC). Prior to analysis unique genotypes that had been scored at <3 microsatellites were removed (n = 8).

Figure 3 Principle component analysis of the individual genotypes obtained from PLEL.

The plot includes PC1 and PC2 of the SNPs and microsatellite genotypes, showing some overlap in the genetics of individuals observed in the two protected areas, Prey Lang (yellow) and Preah Roka/Chhaeb (purple). Prior to analysis unique genotypes that were scored at <4 microsatellites were removed (n = 8). The eigen values for PC1 and PC2 are 0.896 and 0.796, respectively. The variation explained by each principal component is included in brackets on each axis.

Mitochondrial sequencing

A total of 87/163 (53%) of the samples were sequenced at a 152bp sequence of the mitochondrial control region and these sequences were aligned and compared to the 534 available sequences from other range states. A total of 21 haplotypes were observed across the global sample set and clustered into two Asian elephant clades (α and β), and the PLEL elephant samples contained four haplotypes (Fig. 4). One haplotype (I) was in the β-clade and has been observed previously in India, Sri Lanka and Vietnam. This haplotype was only observed in samples collected in the Preah Roka/Chhaeb region. The other three haplotypes were in the α-clade and have previously been observed in other countries (Fig. 4). One of these (IV) was identified in both populations in this study; however, two (II & III) were just observed in samples from the Prey Lang Wildlife sanctuary. The four haplotype sequences identified in the PLEL have been uploaded to NCBI genbank with the following accessions (PP066116–PP066119), but it should be noted that they each align with haplotypes identified from longer mitochondrial sequences previously published as part of a range-wide study.

Figure 4 Haplotype network of a 152bp region of the mitochondrial d-loop.

It uses an alignment of previously published sequences from Vidya, Sukumar & Melnick (2009) and includes the four haplotypes found in the PLEL. The haplotypes are colour coded based on the geographic region the samples were obtained. The two known clades of Asian elephant haplotypes are highlighted in blue (α) and pink (β).

Genetic sexing

Sexing results could be obtained for the majority of unique genotypes. Eight samples were rerun after they showed unclear banding profiles in the initial test. In the final dataset a substantial proportion of the faecal samples were genetically sexed, with 107 of the 163 samples (66%) being classified. A total of 32 out of the 35 unique genotypes were genetically sexed using at least one of their faecal samples. Of these, 15/32 (47%) were female and 17/32 (53%) were male revealing a roughly even sex ratio (1 female: 1.1 males, χ2 = 0.125, p = 0.72, df = 1). No significant difference from an even sex ratio was also found in the two sub populations as tested by a Pearson’s chi-squared test. The 21/22 elephants in Prey Lang Wildlife Sanctuary that were sexed contained eight females and 13 males (1 female: 1.6 males, χ2 = 1.19, p = 0.28, df = 1). The 11/13 elephants in Preah Roka/Chhaeb Wildlife Sanctuaries were seven females and four males (1.75 females: 1 male, χ2 = 0.82, p = 0.37, df = 1).

Genetic diversity

The mitochondrial sequences of the PLEL samples were compared to 13 previously sampled populations in eight countries. They reveal that the two PLEL populations have average levels of haplotype diversity (Prey Lang: 0.55 ± 0.05 and Preah Roka/Chhaeb: 0.46 ± 0.07). However, the two populations in this study differed in their mitochondrial nucleotide diversity. Of note was the nucleotide diversity observed in the Preah Roka/Chhaeb population (0.034 ± 0.005) which was higher than all other sampled populations globally except Sri Lanka (Fig. 5). This is striking given that only two haplotypes were observed in this population; however, they are from highly differing evolutionary histories, one being from the β clade and the other from the α clade (see Fig. 4). This contrasts with the Prey Lang elephants, which only exhibit mitochondrial lineages from the α clade and thus have lower levels of mitochondrial nucleotide diversity (Fig. 4). The nucleotide diversity (microsatellites) did not differ between Prey Lang and Preah Roka/Chhaeb at any of the genetic diversity measures (Table 2).

Figure 5 Genetic diversity calculated for a 152bp region of the mitochondrial d-loop.

It uses the sequences produced in this study and previously published sequences from Vidya, Sukumar & Melnick (2009) for 13 additional wild elephant populations. The two protected areas sampled in this study are highlighted in purple (Preah Roka/Chhaeb) and yellow (Prey Lang). Two diversity measures are shown: haplotype diversity (blue) and nuclear diversity (green) for each population.

Table 2 Genetic diversity calculations using the microsatellite genotyping of individually identified elephants in the Prey Lang Extended Landscape (PLEL).

Only individuals genotyped at ≥4 microsatellites were included. All parameters calculated using hierfstat and standard errors are included in brackets.

Population	N	HObs	HExp	FIS	AR	
Prey Lang	17	0.584 (0.074)	0.627 (0.066)	0.069 (0.105)	3.17 (0.376)	
Preah Roka/Chhaeb	9	0.571 (0.104)	0.639 (0.063)	0.106 (0.134)	2.95 (0.299)	
Note:

N, number of individually identified elephants included; Hobs, Observed heterozygosity; HExp, Expected heterozygosity; FIS, fixation index as calculated in Weir & Cockerham (1984) for small f; AR, Allelic richness.

Population estimate

As there was evidence of movement between Preah Roka and Chhaeb, a combined population estimate was calculated for these two areas. The Prey Lang population estimate was calculated separately. Capwire TIRM results indicate a population estimate of 31 individuals for Prey Lang Wildlife Sanctuary, 95% CI [24–41], and 20 individuals for the Preah Roka–Chhaeb Wildlife Sanctuaries population, 95% CI [13–22].

Habitat models

MaxEnt and fuzzy habitat suitability

Habitat suitability models indicate habitat preferences, mapping potential suitable habitat across the PLEL. The maxent model reached an AUC of 0.941 across the 10 cross-validated models, exceeding the accuracy threshold recommended for application in conservation management. Distance to historic deforestation had the greatest effect on suitability with a 44.7% contribution to model performance, followed by distance to villages (24.4%), and distance to rivers (15.5%). The ensemble Maxent and Fuzzy output identified 276.06 km2 of potentially suitable Asian elephant habitat, the majority of which is found within Prey Lang Wildlife Sanctuary and to the north of Preah Roka Wildlife Sanctuary and Chhaeb Wildlife Sanctuary (Fig. 6).

Figure 6 MaxEnt and weighted habitat suitability ensemble results.

Model results after combining binary maxent and weighted habitat suitability outputs. Image source credits: Esri (2022), IUCN (2021) and UNEP-WCMC (2022).

Connectivity

Connectivity models highlight areas of potentially constricted movement across the landscape. High-cost areas in which movement is restricted are found to the north-west of Prey Lang, south-west of Preah Roka and south-east of Chhaeb wildlife sanctuaries. Similarly, to the south and east of Prey Lang Wildlife Sanctuary. Pinch points demonstrate where landscape connectivity would be disproportionately affected if these pathways were lost are also demonstrated connecting all focus protected areas (Fig. 7). Pinch points are highlighted by corridors of medium connectivity north of Prey Lang Wildlife Sanctuary towards Chhaeb Wildlife Sanctuary. Pinch points are also found within Preah Roka Wildlife Sanctuary as movement is restricted by forest degradation within the sanctuary.

Figure 7 Functional connectivity and connectivity pinch points between core forest.

Functional connectivity and pinch points were modelled in linkage mapper between areas of core forest greater than 25 km2 identified through Morphological Spatial Pattern Analysis. Image source credits: Esri (2022), IUCN (2021) and UNEP-WCMC (2022).

Discussion

This study is the most comprehensive evaluation of a fragmented Asian elephant population in Cambodia. By combining field surveys, genetic techniques and landscape modelling we provide novel insights into the population size, connectivity and habitat usage of this highly threatened ecosystem engineer. Although small and fragmented, the population harbours unusually high levels of mitochondrial genetic diversity, emphasizing the urgency of conservation efforts. Our approach provides a blueprint for assessing similarly threatened populations, as an increasing number of large herbivores face extinction across Southeast Asia (Ripple et al., 2015).

Population size and fragmentation

Using genetic data, we estimated the Asian elephant population in the Prey Lang Extended Landscape, adding to the body of evidence demonstrating the value of non-invasive capture-recapture methods for estimating the size of relatively small populations of elusive wildlife. Understanding population size is key to assess the conservation status of species and to identify management actions needed to secure long-term population persistence (Chaudhary & Oli, 2020). This has been highlighted as increasingly important for Asian elephants, with most population estimates based on educated guesses (Blake & Hedges, 2004). In contrast, our study provides a reliable estimate that can inform management decisions in Cambodia.

The Prey Lang Wildlife Sanctuary elephant population is estimated to number 31 individuals, with a 95% confidence interval of 24 to 41 individuals. The Preah Roka/Chhaeb Wildlife Sanctuaries population is estimated to number 20 individuals, with a 95% confidence interval of 13 to 22 individuals. These estimated population sizes are relatively small but somewhat larger than had been anticipated based on local knowledge gathered to identify survey areas. Minimum viable population sizes in vertebrates depend on factors such as population growth rate (Reed et al., 2003). In the case of elephants, which are severely impacted by habitat fragmentation (Leimgruber et al., 2003; Ripple et al., 2017), it has been proposed that for a population to be viable, it should be >500 elephants (Santiapillai, 1997). This has been suggested to avoid the most severe consequences of inbreeding but does not eliminate them and is unlikely to retain genetic diversity needed for the population to adapt to future challenges, such as climate change or disease (Frankman, Bradshaw & Brook, 2014). Over 1,000 individuals may be necessary on timescales longer than 100 years (Sukumar, 1993) and viability will also depend on the population’s vital rates such as female mortality (De Silva & Leimgruber, 2019). Therefore, the population estimates produced in this study show that the elephants in this landscape are in an extremely precarious situation, with numbers far lower than needed for continued population survival.

The genetic results from this study suggest that the elephants in the landscape are fragmented into two populations. Evidence for the movement of six elephants between Preah Roka and Chhaeb wildlife sanctuaries was produced from the genetic survey, with unique genotypes from the same elephants being found in both protected areas. However, none of the 35 unique genotypes were shared between Prey Lang and the Preah Roka/Chhaeb sanctuaries. This result, combined with the genetic differences observed in the mitochondrial lineages, the population structure analysis and the large barriers to movement observed to the northwest of Prey Lang, suggests that fragmentation into two extremely small populations has occurred recently. These results add to the body of evidence (e.g., Lowe & Allendorf, 2010) on how genetic data, in combination with habitat analyses, can be used to assess fragmentation in wildlife populations. It is therefore of vital importance to (i) increase population sizes as rapidly as possible and (ii) reconnect these populations with each other and/or other neighbouring populations (Johnsingh & Williams, 1999).

This study has identified 27,606 hectares of suitable elephant habitat within the PLEL, the majority of which is within Prey Lang Wildlife Sanctuary. However, as elephants are a long-lived species, with long gestation times, and single calves and can take 13 years to reach sexual maturity (De Silva et al., 2013), there are many constraints that limit rapid population growth. Additionally, potential carrying capacity of this area is difficult to predict as Asian elephant densities vary considerably between previous populations studied, from 0.053 elephants per km2 in China (Sun et al., 2021) to 1.7 elephants per km2 in India (Kumara et al., 2012). Studies performed in populations geographically closest to Cambodia, such as Thailand, have produced densities between 0.1 and 0.7 elephants per km2 (Htet et al., 2021). This would predict that the carrying capacity for the remaining suitable elephant habitat (27,606 ha) could roughly lie between 27–193 elephants. Given this uncertainty we advise that it is a priority to identify any factors that may be limiting population growth (and remove them if possible) and identify whether the area can support a larger population. Even if the population cannot grow rapidly, ensuring that it can grow is vital to retain the genetic diversity that remains.

Another way of maintaining, and potentially increasing, genetic diversity is by reconnecting small fragmented populations with others. This can consist of restoring natural corridors between populations to facilitate movement or by meta-population management via regular human-mediated translocations between populations (Hoffmann, Miller & Weeks, 2020). Given the difficulties and risks (to both elephants and humans) involved in elephant translocations (Santiapillai, 1997) creating natural corridors is often the most suitable. The large separation that now exists between Prey Lang and Preah Roka/Chhaeb also makes this option challenging, and while elephants can thrive in degraded habitat, a significant concerted effort by government and local stakeholders would be required to establish passages across infrastructure barriers and to manage potential human-elephant conflict.

From a methodological standpoint, we encountered high amounts of DNA degradation that needed to be overcome for accurate genotyping, similar to previous attempts to conduct genetic population estimates using faecal samples (Costa et al., 2017; Taberlet et al., 1996). To increase our ability to determine unique genotypes we have combined microsatellite and SNP datasets. Alongside (Crouthers, 2024), this is the first time this has been conducted in an Asian elephant study. We also tested two collection/extraction techniques and showed that samples collected via the swabbing technique produced more suitable DNA for genotyping than the material stored in ethanol. The sample success rate for the SNP genotyping was 65%, which is a substantial improvement on the previous sampling of elephants in a dung survey in the Cardamom Mountains during 2015–2016, where the sample success rate for microsatellite genotyping was 30% (unpublished data). However, this improvement is likely related to the markers being genotyped rather than the collection technique. As in Taberlet, Waits & Luikart (1999), we advise that investigation is conducted to determine the most appropriate sampling, extraction and genotyping technique for a given study, as variation between regions and studies can be high (Piggott Maxine, 2004; Broquet, Ménard & Petit, 2007; Bourgeois et al., 2019).

Genetic diversity and structure

A positive result for both populations considering they are so small was that mitochondrial haplotype diversity were still average when compared to 13 populations in eight other range states (Vidya, Sukumar & Melnick, 2009). Additionally, the nucleotide diversity for Preah Roka/Chhaeb was unusually high, driven by the presence of two highly divergent mitochondrial lineages, one from the α-clade and one from the β-clade (Fernando et al., 2000). These two clades are thought to have evolved independently in the north and south of the Asian elephant range respectively (Fernando et al., 2000; Vidya, Sukumar & Melnick, 2009) but have been found to exist together in some populations, such as those in Sri Lanka and Myanmar (Vidya, Sukumar & Melnick, 2009). Several hypotheses have been proposed to explain their co-occurrence including natural migration and human-mediated transport of elephants, as Sri Lanka was historically a trading hub for captive elephants (Fleischer et al., 2001; Sukumar, 2003). The presence of both haplotype clades in Preah Roka/Chhaeb and not in Prey Lang could be due to random stochasticity in these small populations or increased historical connectedness of the former to other elephant populations (natural or human mediated). Other populations that they could have been naturally connected to would have been across the Thai border or potentially more recently to the remaining strongholds of elephants in Cambodia. Conservation efforts have focused on these strongholds in southern (Cardamom mountains) and eastern (Eastern Plains Landscape) Cambodia (Maltby & Bourchier, 2011). Currently only one scientific article documenting the use of a genetic faecal population estimate has been produced in Cambodia (Gray et al., 2014). It was conducted in 2009 and estimated that Phnom Prich Wildlife Sanctuary, in the Eastern Plains Landscape, contained 136 ±18 (SE) elephants (Gray et al., 2014). In future, it would be useful to gain more regional estimates to put our results within a national context for country-wide conservation planning of this iconic species. Genetic diversity is one of the pillars of biodiversity and the use of genetic diversity in establishing conservation priorities has been strongly supported (DeWoody et al., 2021). We believe our study presents a useful case study where genetic diversity is considered alongside population size and habitat suitability and fragmentation to provide a wholistic understanding of the status of a threatened vertebrate as a basis to inform management and conservation.

Although in this study genetic diversity comparisons with published studies were based on mitochondrial sequences, this study also genotyped nuclear microsatellites that allowed nuclear diversity comparison between the Preah Roka/Chaeb and Prey Lang elephants. No difference in genetic diversity was found, however the ability to compare nuclear diversity across populations at a greater scale throughout Asia would be extremely beneficial, as there are several studies in other range countries (e.g., Zhang et al., 2015; Parida et al., 2022). Microsatellite datasets are notoriously difficult to standardise across studies (Hoffman & Amos, 2005; Moran et al., 2006; Ellis et al., 2011); however, there are multiple strategies that can be used to gain more robust nuclear diversity comparisons in the future. One is to pursue a genotyping method that can be more easily standardised, such as SNP competitive assays (Vignal et al., 2002; Semagn et al., 2014; von Thaden et al., 2017), an alternative approach which alleviates some but not all problems with microsatellite comparisons is the ‘yardstick’ approach (Skrbinšek et al., 2012). This has recently been attempted for Asian elephants using a population in Laos (Budd et al., 2023). This ‘yardstick’ population has been genotyped at the same seven microsatellites analysed in this study. Using the dataset of this population and removing individuals genotyped at <4 of these microsatellites we calculated the rarefied Allelic Richness for the PLEL and the Laos ‘yardstick’ population. The diversity ratio was 0.840 (±0.139 SE). This suggests that the elephants in the PLEL are less diverse than the ‘yardstick’ population, however 88% (n = 28) of populations compared to the ‘yardstick’ have had lower diversity ratios than this one (Budd et al., 2023). This level of nuclear genetic diversity in the PLEL seems to corroborate (Budd et al., 2023)’s findings that elephant populations in Southeast Asia, such as the PLEL, retain unusually high levels of diversity compared to populations elsewhere.

We found an even sex ratio in the samples as a whole; however more males than females were recorded in Prey Lang but a greater number of females compared to male elephants was recorded in Preah Roka/Chhaeb. This could be due to stochasticity in these small populations, but we advise that sex ratios are monitored in any future surveys. A severely skewed female-bias sex ratio can be indicative of elephant populations that have been subjected to high levels of ivory poaching (Sukumar, Ramakrishnan & Santosh, 1998) as males are targeted for their tusks (Mondol, Mailand & Wasser, 2014). This result could therefore indicate a higher poaching threat in the Preah Roka/Chhaeb population compared to the Prey Lang region, however the sample size is too small and would require further investigation before any conclusions are made.

Identification of the relatively high mitochondrial genetic diversity provides some positivity for the elephant population in the PLEL. Typically, populations that have undergone extreme population declines lose substantial genetic diversity, indicating that diversity levels may have been even higher in the past. The remaining high levels of genetic diversity warrant conservation and could aid the prioritisation of the landscape as a national stronghold for this flagship species. However, as genetic diversity is lost more quickly from small populations (Frankham, Ballou & Briscoe, 2010), the conservation need is immediate and requires substantial effort to rescue this remaining diversity.

Habitat suitability and connectivity

Habitat suitability results show extensive areas of suitable Asian elephant habitat in Prey Lang, Preah Roka, and Chhaeb Wildlife Sanctuaries, providing a spatial indication of key areas to focus conservation effort for Asian elephants in the PLEL. However, limited or no recent elephant records exist from some areas modelled as highly suitable habitat, particularly in northern Chhaeb Wildlife Sanctuary on the border with Laos PDR and in central-western Prey Lang Wildlife Sanctuary (Maltby & Bourchier, 2011). Observed discrepancies may be a result of uneven survey effort resulting in an incomplete picture of actual occurrence, and/or limitations in the modelling (Araújo et al., 2019), such as factors not fully accounted for or weighted for, e.g., levels of current or historical human disturbance, access to water sources or seasonal movements. Further research would be required to determine the reasons why occurrence records are concentrated in particular areas of the landscape but apparently absent in other areas with seemingly suitable habitat, so population recovery plans can be better informed. Connectivity results indicate low cost to movement within central regions of protected areas similarly indicating the critical role they play in facilitating species connectivity, and therefore metapopulation sustainability, within the Prey Lang extended landscape. Pinch points in connectivity from northern Prey Lang to Chhaeb Wildlife Sanctuary further demonstrate the importance of what little connectivity there is between the north and south of the PLEL. Currently, this route does require a road crossing. Whilst elephants have been documented to cross roads (Wadey et al., 2018), efforts to decrease resistance to movement in this area through restoration of secondary forests, open habitats and food resources (Wadey et al., 2018) could enhance the viability of road crossing to maintain connectivity. Restoration of forest cover between the two sanctuaries could play a critical role in facilitating the movement of individuals from the two sub-populations identified from the genetic analysis. Despite the areas between Prey Lang and Chhaeb not being designated as protected areas, adequate management could support the re-establishment of vital connectivity (Huang et al., 2019).

This study provides useful insights into areas of suitable habitat and possible connectivity for Asian elephants in the PLEL that will help prioritize conservation efforts. Following on from results of this study, future research should consider the key factors identified here, forest cover, deforestation and proximity to rivers and villages, which align with key variables found in other regions (Huang et al., 2019; Sharma et al., 2020), to develop more locally specific models to improve upon this initial assessment. It is recommended that the relationship between habitat suitability and distance to rivers is explored in more depth to take into account fluctuations during wet and dry seasons which may influence seasonal patterns of suitability and connectivity. This study has shown the importance of forest cover for Asian elephant suitability, and future research could consider generating a bespoke landcover map to overcome the biases and limitations faced in this study using a global forest cover dataset (Shimizu, Ota & Mizoue, 2020). More broadly, our study may serve as a model to demonstrate an integrated approach to threatened species assessments to inform management, combining habitat suitability and connectivity analyses with non-invasive methods to determine population size, genetic diversity and fragmentation.

Conclusions

The Prey Lang Extended Landscape (PLEL) population of Asian elephants represents a critical case study in the conservation of small, fragmented populations of large mammals. Despite its small size (estimated at 51 individuals) and significant threats from inbreeding, human-induced mortality, and habitat fragmentation, the population retains genetic diversity from both major lineages of the species. These findings highlight the dual urgency of addressing immediate threats and leveraging opportunities for recovery within landscapes that still harbor extensive suitable habitat.

The challenges faced by the PLEL elephants are emblematic of broader conservation issues affecting small populations worldwide. Addressing these challenges requires a strategic, scalable approach: (1) stepping up protection of remaining habitat, including through adequate enforcement and management, and utilizing protected area zonation frameworks, with a particular focus on suitable habitat, to avert further degradation as well as snaring pressure; (2) reducing disturbance from human activities, particularly motorized access, logging and other stress-inducing human interventions, in areas within the landscape vital for wildlife survival; (3) maintaining and, where feasible, restoring habitat connectivity within the landscape to support gene flow and increase the resilience of fragmented populations; and (4) targeted research and monitoring to better understand species’ uneven distribution, habitat preferences and threats to inform adaptive management and more effective conservation.

The combined use of genetic analyses and habitat connectivity modelling in this study exemplifies an integrated approach to conservation planning. This methodology not only informs recovery strategies for Asian elephants in Cambodia but also provides a replicable framework for other threatened species facing similar pressures. By demonstrating how genetic diversity and habitat connectivity can be assessed in tandem, this work offers a model for future assessment of other vulnerable populations of Asian elephants and has wider applicability for addressing the challenges of conserving small, isolated populations globally. By prioritizing habitat protection, connectivity, and data-driven management, conservation practitioners can better safeguard biodiversity and enhance the resilience of vulnerable species across diverse ecosystems.

Supplemental Information

Supplemental Information 1 Population size estimate R code.

Supplemental Information 2 Sequence data.

Supplemental Information 3 Comparison of genotyping success of two collection/extraction methods (ethanol and swab) conducted on 20 Asian elephant samples collected in the Prey Lang Extended Landscape.

Genotyping was attempted for 20 Single Nucleotide Polymorphisms (SNPs) via RTPCR amplification.

Supplemental Information 4 Population structure plots for 27 unique genotypes observed in the PLEL.

A priori population sizes for k=2 to k=6 are shown, and each analysis was run in triplicate. The results are consistent between the replicates. Prior to analysis unique genotypes that had been scored at <4 microsatellites were removed (n=8).

Supplemental Information 5 Details of the KASP assays for the 20 SNPs used for individual identification in this study.

Supplemental Information 6 Microsatellite, mitochondrial and sexing primers used in this study.

Supplemental Information 7 Variables and fuzzification parameters used in habitat suitability modelling.

These were selected based on a literature review and guidance from experts consulting on variables likely to impact Asian elephant habitat. An updated forest cover layer was created by reclassifying deforestation events from 2011-2020 as deforested in the 2010 canopy cover dataset. Deforestation hot spots were identified by aggregating forest lost1 in 1 km2 grid events, using the create space-time cube and emerging hot spots tool in ArcGIS Pro 2.8 to evaluate spatio-temporal trends. The analysis was run for historical deforestation (2001 to 2015) and deforestation inclusive of recent events (2001 to 2020). Euclidean distance was applied to recent and historic deforestation hotspots, defined as new, consecutive, intensifying and persistent hot spots, village and river datasets. The deforestation, village and road density variables were used to represent anthropogenic disturbance in the PLEL. Fuzzification transformations were applied to variables for the weighted fuzzy model. Hansen, M. C., P. V. Potapov, R. Moore, M. Hancher, S. A. Turubanova, A. Tyukavina, D. Thau, S. V. Stehman, S. J. Goetz, T. R. Loveland, A. Kommareddy, A. Egorov, L. Chini, C. O. Justice, and J. R. G. Townshend. 2013. “High-Resolution Global Maps of 21st-Century Forest Cover Change.” Science 342 (15 November): 850–53. Data available from: https://glad.umd.edu/dataset/global-2010-tree-cover-30-m.

Supplemental Information 8 Resistance values for variables included within functional connectivity models.

Supplemental Information 9 Summary of the 35 unique genotypes identified in the sample set.

Supplemental Information 10 Genetic study methods.

Mr. Uk On Norong facilitated the use of the Royal University of Phnom Penh Conservation Genetics Laboratory and Seanghun Meas provided management support. The Forestry Administration, Fauna & Flora International, Wildlife Conservation Society, Conservation International and Wild Earth Allies contributed occurrence records. Hang Chandaravuth, Phoeun Chhunheang, Kong Borey, Yim Raksmey, Hun Seiha and Song Det participated in field surveys. Ratha Prum, Sopheap Yun and Muhammad Ghazali contributed to the genetic analysis work. We thank Helen Senn for advice and preliminary work.

Additional Information and Declarations

Competing Interests

The authors declare that they have no competing interests.

Author Contributions

Pablo Sinovas conceived and designed the experiments, performed the experiments, analyzed the data, authored or reviewed drafts of the article, and approved the final draft.

Chelsea Smith performed the experiments, analyzed the data, prepared figures and/or tables, authored or reviewed drafts of the article, and approved the final draft.

Sophorn Keath performed the experiments, authored or reviewed drafts of the article, and approved the final draft.

Nasak Chantha performed the experiments, authored or reviewed drafts of the article, and approved the final draft.

Jennifer Kaden performed the experiments, analyzed the data, authored or reviewed drafts of the article, and approved the final draft.

Saveng Ith performed the experiments, authored or reviewed drafts of the article, and approved the final draft.

Alex Ball conceived and designed the experiments, performed the experiments, analyzed the data, prepared figures and/or tables, authored or reviewed drafts of the article, and approved the final draft.

Field Study Permissions

The following information was supplied relating to field study approvals (i.e., approving body and any reference numbers):

All field work was authorised by the Cambodian Ministry of Environment (under USAID Greening Prey Lang Project).

DNA Deposition

The following information was supplied regarding the deposition of DNA sequences:

The four haplotype sequences are available at NCBI GenBank: PP066116 to PP066119.

Data Availability

The following information was supplied regarding data availability:

The R code used to estimate population size is available in the Supplemental File.

The raw microsatellite fragment analysis files and SNP fluorescent data are available at Dryad:

- Ball, Alexander; Sinovas, Pablo; Smith, Chelsea et al. (2025). Asian elephant Microsatellite raw fragment files produced from faecal samples collected in the Prey Lang Extended Landscape, Cambodia [Dataset]. Dryad. https://doi.org/10.5061/dryad.m905qfv88

- Ball, Alexander; Sinovas, Pablo; Smith, Chelsea et al. (2025). SNP genotyping raw flourescent values from Asian Elephant in Prey Lang Extended Landscape, Cambodia [Dataset]. Dryad. https://doi.org/10.5061/dryad.ttdz08m5g.

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
