# Peer review of "Giants in the landscape: status, genetic diversity, habitat suitability and conservation implications for a fragmented Asian elephant (Elephas maximus) population in Cambodia"

_PeerJ, doi:10.7717/peerj.18932_

## Round 0.1 · original submission · Major Revisions

Thank you for submitting this interesting research to PeerJ. I regret that I am unable to accept the manuscript for publication, at least in its present form. However, I am prepared to consider a new version that very carefully considers the many concerns highlighted by the reviewers. These need to be addressed comprehensively in a new version. Such a revised manuscript is likely to be reviewed again and there is no guarantee of acceptance. When you revise the manuscript, please prepare a detailed explanation about how you have dealt with all the reviewer comments, as well as my own ones.

Overall, the standard of preparation and writing needs to be greatly improved. For example, basic paragraph structure should be checked and improved throughout, e.g. lines 47-58, 87-93, 113-116 (and please check the rest of the manuscript). See further advice in this published paper, which you will hopefully find useful: The art of writing science - https://onlinelibrary.wiley.com/doi/full/10.1002/pro.514
"The first sentence of each paragraph should tell the reader what you expect them to get out of the paragraph that follows, which makes their job of following it far easier."

The basic spacing of the text in the manuscript varies from section to section. Please fix it.

Lines 11-115. Please clarify exactly what interviews were done, and how many were carried out. What formal permission or ethical approval did you have in order to conduct interviews with people?
Line 305. "Both methods". Please state clearly what methods were used. See the advice on paragraphs in: "The art of writing science"
Line 329-332. Is this meant to be a paragraph?
L363-370. Please fix the paragraph structure.
Line 487. What results? Try to be more specific and descriptive.
Line 487-491. This is not a paragraph and requires substantial editing.
L493-504, L520-530. This is Discusion but is currently lacking any references.

Reviewer 3 is correct - the overall focus of the manuscript if far too narrow (given the topics being studied) and needs to be expanded in order to appeal more broadly to more readers.

Delete this text: We will discuss each of these points in turn.

Reviewer 1 ·

Basic reporting

This article is clearly and well written and easy to follow.

Experimental design

The study is well-conducted and well-articulated; however, it lacks detail and clarity regarding the use of replicates in the molecular work. In non-invasive genetics, it is standard practice to replicate data to mitigate potential errors arising from low-quality and low-quantity DNA templates. Gender identification is typically carried out in duplicate, and microsatellite data analysis is conducted at a minimum in duplicate, ideally in triplicate. It remains unclear whether SNP genotyping was replicated for reliability. Additionally, the use of blanks or negative controls to monitor contamination risks is not mentioned. Although two genotyping methods were employed, potentially reducing error chances, some reassurance for the reader on this aspect would be beneficial.

Regarding analysis, several methods, including Micro-Checker, can detect genotyping data errors. More details on how this was executed or the precautions taken with the data would be valuable.

I think it would be useful to also include some basic descriptive statistics from the genoptyes e.g. levels of expected and observed heterozygosity, FIS, number of alleles etc. This would be useful for future studies in the region too and may be interesting to analyse between the two clusters identified.

It would also be beneficial to mention briefly in the text which primers or sources were used for the mtDNA analysis. Since this aspect of the study is novel and the details have not been previously published or verified, including these directly in the manuscript, rather than in an appendix, would be helpful for readers. The sentence, "Two primer pairs that each amplify a roughly 100bp fragment in an overlapping region of the control region were used," is unclear. Clarifying the rationale behind this choice, possibly due to the challenges of amplifying mtDNA from degraded or poor-quality DNA, would enhance understanding. However, the supplementary material lacks further details on how the sequences were sequenced, which is a concern in my opinion, especially given the difficulty in sequencing amplicons below 100 bp. Lastly, information on which sequences or Accession codes were used to design the primers would be valuable.

Validity of the findings

Overall a very interesting study with many elements included in the study. Apart from my comments above related to the experimental design, the data is well analysed and explained.

Reviewer 2 ·

Basic reporting

All good

Experimental design

The experimental design seems sound and appropriate. Please note that my knowledge of genetics is quite general and might have missed technical problems. As far as I can tell, the methodology is adequate.

Validity of the findings

Same as above. All seems good.

Additional comments

Congratulations for the good an important work for elephant conservation in SE Asia.
I have two very minor comments:
1. Line 308 -- it should be PCA, not PVA
2. Line 368 and others -- for this spatial scale I recommend using sq-km, rather than Ha

Reviewer 3 ·

Basic reporting

I appreciate the amount of work (data collection) carried out in this study, but the reporting needs to be of sufficient scientific quality, which, in the manuscript’s current form, I am afraid is not present. See my feedback below, which may help improve the manuscript to some extent. However, I believe the manuscript requires significant revision before being considered for publication.

Comments on Abstract:
The abstract starts well, but the sentences in the latter part need to be better connected. Please do not use paragraphs in an abstract. Also, in lines 24-25, the authors report their results without carefully stating the goals and methods of the study. Please revise and make it more coherent for readers. See below for other comments -
Line 30: Since you first introduced the three places in a specific order, maintain the same order throughout the manuscript. Please change “Chhaeb and Preah Roka” to “Preah Roka and Chhaeb” here.
Lines 31-35: Please maintain the order of reporting. First, Preah Roka and Chhaeb (i.e., the northern population), and third, Prey Lang (i.e., the Southern population).
Lines 33-34: “estimated to number” in both sentences must be rephrased. Use something like “the estimated population size was...”
Line 36: ...protected areas; however,...
Line 41-44: Please check sentence structure.

Comments on Introduction:
The introduction is very short and should be substantially expanded. The authors started talking about the current study population (cf. Line 54) just after an initial few sentences, limiting the bigger scope. Please start with the ecological importance of conserving animals, current challenges, limitations, etc. Then, explain how, in general, previous studies have dealt with these various concepts. Afterwards, Elephants should be introduced in the picture, along with your current study locations and existing challenges, like deforestation, habitat loss, etc. A major concern of the introduction is its restriction to Elephants, making the scope of the work extremely narrow.
Line 61: Please discuss the benefits.
Line 71-83: This paragraph is relatively better written than the others in the introduction.
It isn't easy to provide more specific comments to the introduction until it is substantially revised and expanded.

Comments on Discussion:
I find the discussion section to be much better than the other sections of the manuscript. However, I still feel that the scope is extremely narrow, and it should at least be broadened to some amount. See below for some comments -
Line 385: The authors used “first time” for the third time in this manuscript. Please remember that more or less all original research is performed for the first time, or where the findings are ‘novel’. I would suggest using these words repeatedly, as they don’t carry much weight.
Lines 388-389: Please mention the basis of this anticipation.
Lines 395-397: It seems like these sentences have been copied from somewhere else and then pasted.
Line 520: Use of “first” for the fourth time! See my concern above.

Experimental design

Comments on Methods and Results:

These sections need considerable expansion in detail. It is impossible to assess how systematic the data collection method was. No information about the study areas, like coordinates or maps, is provided. My expertise in genetics is limited, so I can’t comment on them. But see below for specific comments on other areas -
Lines 87-90: Please provide coordinates of the area. Also, is there any reference to dry and wet seasons? Like weather data?
Line 92: What are recent years?
Lines 113-114: What kind of interviews? Structured? Semi-structured? Did you have ethical approval to conduct the interviews?
Lines 115-116: Please add sufficient information about these areas (like a map showing the boundaries).
Line 117-118: Who are the trained researchers? Authors of this study or other people as well?
Line 119: replace ‘genetic’ with ‘faecal’ samples.
Lines 167-168: Please add details on the PCA.
Line 225 231: This is extremely vague. No idea how authors measured these parameters. No references were given as well.
Lines 245-246: What is the local expert’s opinion? Please be precise and avoid using these uninformative phrases in scientific writing.
Lines 272-274: This should be part of your methods, not results.
Line 308: PCA instead of PVA. Where are the PCA details? Eigenvalues of the PCs? % of variation explained by the PCs? Rotation details? They must be reported.
Lines 336-338: The results/differences are not even close to statistical significance. How do you say they are ‘higher’/lower?

Validity of the findings

Supplementary files are adequate. Genotype data are provided. R Script is provided.

Additional comments

References and Figures:
There are plenty of formatting issues with the references. Some of them do not have DOIs; some have years in brackets, and some do not; in some cases, the last author’s name starts after “&”, and in some, after “,”. I will not point out every detail; please check and correct them carefully.
Figure 2: Two eigenvalues of the corresponding PCs are highlighted, but what are the actual values?
Figure 6: Scale missing?

---

## Round 0.2 · Major Revisions

Thank you for carrying our detailed revisions to the manuscript. From my own reading, and from the reviewer comments, it is clear that the manuscript has greatly improved. Nevertheless, there is still room for further improvements to be made. As discussed via email, please insert some clear details in the Methods, regarding ethics and permits for the research. Remove future tense from the Introduction (last paragraph). The results and presentation of results (see reviewer comments) need to be improved. L457-this text currently make little sense, so please revise it. L472-instead of "They", describe exactly what you mean, as it is currently not clear. The focus of the Discussion (on Asian elephants) is too narrow. Overall, paragraph structure could still be improved throughout manuscript. For example, in the Discussion, consider using the following structure for each paragraph: "state your case" and "make your case". State Your Case = one or two sentences at the start of the paragraph about what you found. Make Your Case = use the rest of the paragraph to highlight why your results are important/interesting, in light of previous research (cited references). I look forward to receiving your fully revised manuscript.

Reviewer 1 ·

Basic reporting

N/A

Experimental design

n/a

Validity of the findings

n/a

Additional comments

I am satisifed that the previous comments I made to the MS have been addressed. However, despite descriptive population stats being included in Table 1 and broadly discussed in the discussion, they have not been specifically mentioned in the text. I think it would also be useful to provide some average or population level descriptions of genetic diversity for the study, and then discuss how these values broadly compare to other elephant studies.

Were significance levels for the FIS vlaues also calculated? Those details could also be provided. Some discussion as to what they mean in the context of this study would also be useful.

Finally, legend for Table 1 should also include all the abbreviations.

Reviewer 3 ·

Basic reporting

I thank the authors for improving the manuscript based on the reviewers' comments. However, I have one major concern regarding the PCA findings and the analysis conducted later based on that. See below for specific comments -

Minor comments:

The abstract improved considerably and quite clear now.

Introduction: Great job! Some comments -

Line 55 - You can consider adding some examples after '...herbivores' (such as Asian Elephants and Buffalo).
Line 70-72 - Put the study 'Zakaria et al. 2024' as a regular main text citation, and re-write the sentence accordingly.
Line 115 - A general tip. You are writing the paper after you have conducted the study. So, instead of 'will use', use 'we used'. Example: In this study, we used a non-invasive genetic sampling survey to assess population size, sex ratio and genetic diversity of the PLEL elephant population. We also used modelling approaches to identify areas of suitable habitat and key connectivity corridors for the species.
Line 119 - the methodology 'may' provide as a blueprint.

Discussion:

Line 468-470: 'with Sukumar estimating'? Where is the reference? 'de Silva & ... noting'. Please DO NOT write sentences like this. Simply mention the finding/statement and report the study as regular main text citations. Example: An estimated 1000 or over individuals...(Ref).

Line 597 - This study provides...

Figures:

There are major issues with figures 1,6, and 7. I think it's due to their formats? Scales and legends are missing. In Figure 6 and 7, the font style, appearance, and color need to be changed.

Major comment:

Results:

Line 374-383: Now that the PCA details are provided, I can see that they are not robust. Total variance explained by the two PCs is <30%, and the eigenvalues of the PCs are not even 1. This potentially suggests that there are no distinct clusters present. But then I don't get why you conducted separate analysis for the two populations? Please explain.

Experimental design

NA

Validity of the findings

NA

Additional comments

NA

---

## Round 0.3 · Major Revisions

Thank you for carrying out some of the required revisions. Please carefully read the comments of both the reviewers. You will find that neither of them are satisfied that previous suggested edits have been adequately carried out. We have now seen three versions of this study, and I am willing to allow one final attempt to make the required changes, and to bring the manuscript up to the required standard. Please note that if you do not carry out the changes, you risk having the next version of this manuscript rejected.

Introduction, L55. Latin names for species are should be given once; the first time they are mentioned.

An "in prep" paper cannot be used to support key aspects of the research [Crouthers, R.C., Linklater, W., Ritchie, P., In, V., Sieng, D., Ball, A. In Prep. Genetic structure, diversity and regional importance of Cambodia's largest elephant population]. Note also the formatting error in the reference list.

L227. Fix the spacing error (check the whole manuscript carefully for minor issues).

L438. Fix the formatting/alignment of the text.

With only two sentences, the opening paragraph of the Discussion is quite limited and weak. First, in the Discussion, highlight the major findings in the opening paragraph. And then discuss more specific findings in subsequent paragraphs.

The reviewers and I are really trying to help with the quality of the final version of this manuscript, so please use our advice.

Reviewer 1 ·

Basic reporting

N/A

Experimental design

N/A

Validity of the findings

n/a

Additional comments

I believe the authors have addressed all comments to the best of their ability. For studies that use nuclear markers and microsatellites to assess genetic diversity, it is entirely reasonable to report these findings alongside those of similar studies. While these studies may use different markers than those in your own research, providing this context is valuable. You may wish to include a caveat noting these methodological differences, but you are not being asked to make direct comparisons across labs—simply to discuss the findings with any necessary qualifications. It is in my opinion better to acknowedge these studies rather than ignore them.

Reviewer 3 ·

Basic reporting

Thanks, authors, for revising the manuscript. I will only comment on my earlier question regarding the PCA analysis. Thanks for sharing the interesting article that suggests PCA-based findings in population genetics need careful evaluation. However, I still have the same comment regarding the interpretation of PCA results:

Wrt. Lines 381-385:

1. It is not robust (~30% variance of the data explained).
2. Eigenvalues are not above '1'. So, they certainly do not suggest a 'small degree of genetic divergence'.

In summary, authors should not rely on a method that is not applicable to their data and then ambiguously interpret the results in their favour.

Experimental design

NA

Validity of the findings

NA

Additional comments

In my first round of review, I highlighted that I am not a genetics expert. So, I believe that other reviewers, who are genetics experts, have evaluated the manuscript too.

---

## Round 0.4 · Minor Revisions

Please remove all the references from the Results section, and move the related sections of methods and discussion text, to their respective sections. See Hall 2022, Crouthers 2024, Vidya et al. 2009, Hoffman and Amos 2005, as well as the other references in the Results. Results section = Results text.

---

## Round 0.5 · accepted · Accept

Thank you for the final edits.